# Differentiation Trajectory of Limbal Stem and Progenitor Cells under Normal Homeostasis and upon Corneal Wounding

**DOI:** 10.3390/cells11131983

**Published:** 2022-06-21

**Authors:** Zhenwei Song, Brian Chen, Chi-Hao Tsai, Di Wu, Emily Liu, Isha Sharday Hawkins, Andrew Phan, James Todd Auman, Yazhong Tao, Hua Mei

**Affiliations:** 1Department of Ophthalmology, School of Medicine, The University of North Carolina at Chapel Hill, Chapel Hill, NC 27599, USA; zhenwei_song@med.unc.edu (Z.S.); chtsai@med.unc.edu (C.-H.T.); emilyliu@email.unc.edu (E.L.); iisha@live.unc.edu (I.S.H.); 2School of Medicine, Hunan Normal University, 371 Tongzipo Road, Changsha 410081, China; 3Department of Biostatistics, Gillings School of Global Public Health, The University of North Carolina at Chapel Hill, Chapel Hill, NC 27599, USA; brichen@live.unc.edu (B.C.); dwu@unc.edu (D.W.); 4Division of Oral and Craniofacial Health Research, Adams School of Dentistry, The University of North Carolina at Chapel Hill, Chapel Hill, NC 27599, USA; 5Department of Psychology and Neuroscience, School of Medicine, The University of North Carolina at Chapel Hill, Chapel Hill, NC 27599, USA; aphan@unc.edu; 6Department of Pathology and Laboratory Medicine, School of Medicine, The University of North Carolina at Chapel Hill, Chapel Hill, NC 27599, USA; jtauman@email.unc.edu (J.T.A.); yt185@duke.edu (Y.T.); 7Department of Cell Biology and Physiology, School of Medicine, The University of North Carolina at Chapel Hill, Chapel Hill, NC 27599, USA

**Keywords:** hedging, transaction costs, dynamic programming, risk management, post-decision state variable

## Abstract

Limbal stem cells (LSCs) reside discretely at limbus surrounded by niche cells and progenitor cells. The aim of this study is to identify the heterogeneous cell populations at limbus under normal homeostasis and upon wounding using single-cell RNA sequencing in a mouse model. Two putative LSC types were identified which showed a differentiation trajectory into limbal progenitor cell (LPC) types under normal homeostasis and during wound healing. They were designated as “putative active LSCs” and “putative quiescent LSCs”, respectively, because the former type actively divided upon wounding while the later type stayed at a quiescent status upon wounding. The “putative quiescent LSCs” might contribute to a barrier function due to their characteristic markers regulating vascular and epithelial barrier and growth. Different types of LPCs at different proliferative statuses were identified in unwounded and wounded corneas with distinctive markers. Four maturation markers (*Aldh3*, *Slurp1*, *Tkt*, and *Krt12*) were screened out for corneal epithelium, which showed an increased expression along the differentiation trajectory during corneal epithelial maturation. In conclusion, our study identified two different types of putative LSCs and several types of putative LPCs under normal homeostasis and upon wounding, which will facilitate the understanding of corneal epithelial regeneration and wound healing.

## 1. Introduction

The corneal epithelium is under constant renewal throughout life, which is supported by limbal stem cells (LSCs). Limbal stem cells are located at the basal limbal epithelium [1,2] and are thought to represent less than 1% of limbal epithelial cells [3,4,5]. Considerable effort has been devoted to the study of LSC-enriched populations (e.g., limbal epithelial cells, side populations of limbal epithelial cells, and slow-cycling limbal epithelial cells) and a list of stem/progenitor markers has been screened out, including ΔNp63α, ABCG2, ABCB5, FZD7, KRT14, and N-cadherin [6,7,8,9,10,11,12,13,14,15]. However, none of the markers can distinguish LSCs from the surrounding progenitor and niche cells. Recently, due to the development of the technology of single-cell RNA sequencing and single-cell quantitative real-time PCR, researchers were able to analyze limbal epithelial cells at the single-cell level. Many studies were performed on human corneas from healthy and keratoconus samples and suggested more than one clusters (heterogeneity) as putative limbal stem/progenitor cells with characteristic markers [16,17,18,19]. The studies using human samples provided more human/clinical related data but have the disadvantages of significant sample variation due to the non-uniform genetic background of different donors, which makes it difficult to compare the data from different donors (e.g., among healthy corneas, between healthy corneas and pathological corneas). The single-cell studies were also performed in animal models, including rabbits and mice [20,21,22]. Similar to the data obtained using human samples, LSCs were found to be heterogeneous in rabbits and mice [20,21,22]. More specifically, researchers discovered the existence of two distinctive LSC populations located at the “outer” and “inner” parts of the limbus, respectively [22]. The “outer” LSCs (cluster 3 with markers *Gpha2*, *Cd63*, and *Ifitm3*) had less proliferative cells under normal homeostasis and expressed higher levels of *Krt15* and *Krt14* and a lower level of *Krt12* than the “inner” LSCs (cluster 4 with markers *Atf3* and *Mt1-2*), suggesting that the “outer” LSCs represent a more undifferentiated and quiescent stem cell population.

In this study, we explored the heterogeneity of limbal epithelial stem and progenitor cells by analyzing the label-retaining cells from H2B-GFP mice using single-cell RNA sequencing under normal homeostasis and wound healing.

## 2. Methods

### 2.1. Mice

The R26-M2rtTA; TetOP-H2B-GFP transgenic mice (stock No.: 016836, the Jackson Laboratory, Bar Harbor, ME, USA) were used in this study. The experiments were conducted under the protocol approved by the Institutional Animal Care and Use Committee (IACUC) at the University of North Carolina at Chapel Hill and were in compliance with the Association for Research in Vision and Ophthalmology Statement for the Use of Animals in Ophthalmic and Vision Research.

### 2.2. Pulse-Chase Experiment and Wound-Healing Procedure

During the pulse phase, the 4-week-old mice (approximately equal numbers of male and female mice) were fed with doxycycline (2 mg/mL) in sucrose (5%, *w/v*) water for 4 weeks, followed by a 4-week chase phase during which the corneas underwent two rounds of wound healing procedures (timeline shown in Figure 1). During the wound-healing procedure, the mice were under deep anesthesia, and 1% tetracaine hydrochloride (Oceanside Pharmaceuticals, Bridgewater, NJ, USA) was applied on the OD eyes. Absence of sensation was confirmed by absence of corneal reflex before surgery. Then, 5% povidone iodine (RICCA Chemical Company, Arlington, TX, USA) was applied as the preoperative topical antiseptic on the ocular surface and surrounding area, and removed by sterile Q-tips (COPAN Diagnostics, Murrieta, CA, USA). The central corneal epithelium was carefully removed by Algerbrush (Ambler Surgical, Exton, PA, USA), and the size of the wound was confirmed by fluorescein staining (Sigma-Aldrich, St. Louis, MO, USA). Wounded eyes were treated with BNP ointment (vetropolycin, Bausch & Lomb, Laval, QC, Canada) post-surgery to prevent bacterial infection.

### 2.3. Hematoxylin and Eosin (H&E) Stain

The enucleated eyes were embedded in optimal cutting temperature medium (OCT) (Fisher Healthcare, Houston, TX, USA) and cross-sectioned at 10 µm thickness using a cryostat (Thermo Fisher Scientific, Waltham, MA, USA). Eye sections were fixed with neutral 10% formalin (Sigma Aldrich, Meick KGaA, Darmstadt, Germany) for 10 min at room temperature followed by a standard Hematoxylin and eosin stain (Thermo Fisher Scientific, Waltham, MA, USA) following the manufacturer’s protocol. The sections were mounted in mounting medium (Fisher, Thermo Fisher Scientific, Waltham, MA, USA) for microscopic study.

### 2.4. Co-Localization Study of the Label-Retaining Cells with the K14^+^ and p63-Bright Cells

Cryosections of mouse eyes were fixed by 1% paraformaldehyde (PFA, Electron Microscopy Sciences, Hatfield, PA, USA) for 15 min at room temperature, washed in phosphate buffered saline (PBS, Sigma Aldrich, Meick KGaA, Darmstadt, Germany) 4 times, and incubated in blocking buffer (PBS containing 10% goat serum (Sigma Aldrich, Meick KGaA, Darmstadt, Germany), 1% bovine serum albumin (BSA, Thermo Fisher Scientific, Waltham, MA, USA), and 0.4% Triton X-100 (Sigma Aldrich, Meick KGaA, Darmstadt, Germany) for 1 h at room temperature. The cells were incubated with primary antibody (K14: MA5-11599, 1:500 dilution, Thermo Fisher Scientific, Waltham, MA, USA; p63: 13109S, 1:200 dilution, Cell Signaling Technology, Danvers, MA, USA) in blocking buffer for 2 h at room temperature. Excess antibody with non-specific binding was washed away by PBST (PBS containing 0.025% Triton X-100). Secondary antibodies (Alexa Fluor 488 goat anti mouse, ab150117, 1:1000 dilution, Abcam, Cambridge, MA, USA; Alexa Fluor 555 goat anti rabbit, A21428, 1:1000 dilution, Thermo Fisher, Waltham, MA, USA) were used in blocking buffer for 1h at room temperature, followed by PBST wash 3 times. Nuclei were counterstained by Hoechst 33342 (3570, Thermo Fisher Scientific, Inc., Waltham, MA, USA). Images were taken by Olympus IX81 (Olympus, Center Valley, PA, USA).

### 2.5. Cell Isolation and Fluorescence-Activated Cell Sorting (FACS) to Obtain the Label-Retaining GFP+ Cells

Limbal epithelial cells were collected following a previously optimized protocol [23] (“Method 4” in the paper). In brief, after euthanasia, the corneal side of limbal epithelial cells was marked by gently pressing a 1.5 mm trephine on the central cornea. Then, the whole eye globes were dissected and cleaned by removing the excess conjunctiva (the remaining conjunctiva was within 1 mm zone from the limbus), muscle, and fat using a spring scissor. The eye globes were incubated in 2.4IU/mL Dispase II (Sigma-Aldrich, Merck KGaA, Darmstadt, Germany) for 2 h at 37 °C followed by an additional digestion in 1mg/mL collagenase A (Sigma-Aldrich, Merck KGaA, Darmstadt, Germany) for 20 min. The limbal epithelial sheet was gently scraped off from the globe using curved-tip forceps, followed by an incubation in 0.25% Trypsin/EDTA (Thermo Fisher Scientific, Waltham, MA, USA) at 37 °C for 8 min to obtain the single-cell suspension. Cell suspension was pipetted through a 30-gauge needle to facilitate the generation of a single-cell suspension for fluorescence-activated cell sorting (FACS). Limbal epithelial cells from wild type C57Bl/6J mice were included as the negative control. The GFP+ cells were sorted into Dulbecco’s Modified Eagle Medium: Nutrient Mixture F-12 (DMEM/F12) (Gibco, Waltham, MA, USA) containing 5% fetal bovine serum (FBS, Gibco, Waltham, MA, USA), 1% penicillin/streptomycin (Gibco, Waltham, MA, USA), and 10 µg/mL gentamicin/0.25 µg/ml amphotericin (Thermo Fisher Scientific, Waltham, MA, USA) for single-cell RNA sequencing.

### 2.6. Single-Cell RNA Sequencing and Data Analysis

Single-cell RNA sequencing was performed using the 10x Genomics following the manufacturer’s protocol. In brief, the GFP^+^ cells from unwounded intact eyes and from eyes that underwent wound-healing procedures were collected from FACS and immediately loaded onto the 10× Chromium Controller (Chromium Single Cell 3′ Library & Gel Bead Kit v3) (10× Genomics, Pleasanton, CA, USA), followed by PCR amplification and sequencing on NextSeq 500 high output (Illumina, San Diego, CA, USA). The sequencing data obtained from the NextSeq500 were demultiplexed using TGL analysis servers and the resulting FASTQ files were analyzed using the R version 3.6.0 of Seurat 2.3.4, which uses canonical correlation analysis to identify shared correlation structures and can visualize using dimension reduction procedures including PCA and t-SNE using 10 dimensions. Initial data screening for quality control was performed to exclude cells with low reads, low detected genes, and high apoptotic genes. Seurat’s analysis packages were used to perform unsupervised clustering of the GFP^+^ cells to sort them into different cell clusters based on the similarity of gene expression patterns. The clusters from both samples (unwounded eyes and eyes that underwent wound-healing procedures) were pooled together for the differentiation trajectory analysis.

### 2.7. Immunohistochemistry of Putative LSC Markers

Human corneas were a generous gift from the eye bank “Miracles In Sight” (a non-profit eye bank). The corneas were embedded in OCT (Fisher Healthcare, Waltham, MA, USA) and cryosectioned at 10 µm thickness. The sections were fixed in 1% paraformaldehyde (Electron Microscopy Sciences) in PBS (Sigma Aldrich) for 10 min at room temperature, incubated in 0.3% Triton–X100 (Sigma Aldrich, St. Louis, MO, USA) in PBS (Sigma Aldrich) for 10 min, blocked with 5% BSA (Thermo Fisher Scientific) in PBS (Sigma Aldrich, St. Louis, MO, USA) for 30 min, and incubated with primary antibody (FMO2, 67019-1-IG, 1:200 dilution, Proteintech, Rosemont, IL; EDN2, PA3-002, 1:1000 dilution, Thermo Fisher Scientific, Waltham, MA, USA; ADM, 10778-1-AP, 1:100 dilution, Proteintech, Rosemont, IL, USA) in PBS (Sigma Aldrich) containing 1% BSA (Thermo Fisher Scientific) overnight at 4 °C. After washing the excess antibody with PBS (Sigma Aldrich), the sections were incubated in secondary antibody (ab96883, 1:1000 dilution, Abcam, Cambridge, United Kingdom; A11012, 1:1000 dilution, Thermo Fisher Scientific) for 1 h and washed with PBS (Sigma Aldrich), followed by nuclear counterstain with Hoechst 33342 (MillporeSigma, Burlington, MA, USA). Images were taken by Olympus IX81 (Olympus, Center Valley, PA, USA).

### 2.8. Corneal and Limbal Epithelial Isolation and Quantitative Real-Time PCR

Corneal and limbal epithelial cells were isolated from human corneas following our previous protocol [10]. In brief, human corneas were cleaned by removing the residue blood, iris, corneal endothelium, excess conjunctiva (the remaining conjunctiva was within 2 mm zone from the limbus), and excess Tenon’s capsules, followed by a firm press using an 8 mm-diameter trephine to separate the cornea from the limbus. Corneal button and limbus were incubated in separate tubes containing 2.4 U/mL Dispase II (Sigma-Aldrich) in medium (DMEM/F12 (Gibco) with 5% FBS (Gibco), penicillin–streptomycin (Gibco), and gentamicin/amphotericin B (Thermo Fisher Scientific) at 37 °C for 2 h. The corneal and limbal epithelium were obtained from the corneal button and limbus, respectively, by mechanical scraping with care. RNAs were extracted from the corneal and limbal epithelium using RNeasy Micro Kit (Qiagen, Hilden, Germany) following the manufacturer’s protocol, followed by DNase treatment (Thermo Fisher Scientific, Waltham, MA, USA) and reverse transcription using High Capacity cDNA reverse transcription kit (Applied Biosystems, Waltham, MA, USA). Quantitative real-time PCR was performed to examine the expression of ten genes selected from the single-cell RNA sequencing data as putative maturation markers (*Mgarp*, *Aldh3*, *Slurp1*, *Tkt*, *Lypd2*, *Aqp5*, *Dbx2*, *Spink7*, *Piezo2*, and *Krt12*) using the applied biosystems StepOnePlus PCR system and PowerUp™ SYBR^®^ Green Master mix (Applied Biosystems, Waltham, MA, USA) following the standard fast cycling mode (UDG activation at 50 °C for 2min and Dual-Lock^TM^ DNA polymerase at 95 °C for 2 min, followed by 40 cycles of amplification including denaturing at 95 °C for 15 s and annealing/extension at 60 °C for 1 min) and the default dissociation step (95 °C for 15 s, 60 °C for 1 min, and 95 °C for 15 s). The primers used to detect the ten genes were shown in Table 1.

### 2.9. Statistical Analysis

All data points were collected from 4–6 independent experiments. One-way ANOVA and Students’ *t*-test were used for multiple comparison and comparison between two specific groups. *p* < 0.05 was considered statistically significant.

## 3. Results

### 3.1. Validation of the Label-Retaining Transgenic Mouse Model for the Study of LSCs

The H2B–GFP transgenic mice were employed in this study to enrich LSCs in the label-retaining (GFP^+^) cells. Under doxycycline induction, cells were labeled with GFP in a widespread pattern (Figure 2A), which facilitated the inclusion of all LSCs whose expression profile was still unknown. Before performing the LSC study using the H2B–GFP mice, we examined whether this mouse model was suitable to study LSCs from three aspects: (1) whether there were false negative GFP signals with doxycycline induction; (2) whether there were false positive GFP signals without doxycycline induction; (3) whether the label-retaining cells contained LSCs.

The H2B–GFP mice were fed with doxycycline (2 mg/mL) in sucrose water for 4 weeks during the pulse phase, followed by 2 rounds of a corneal epithelial wound-healing procedure during the subsequent 4-week chase phase (Figure 1). First, we examined whether our targeted tissue (limbal epithelium) was labeled with GFP upon doxycycline induction. Data showed that both corneal and limbal epithelial cells expressed strong GFP after induction at the end of the pulse phase (Figure 2A–C), indicating that there was no false negative signal in our targeted tissue. Corneal and limbal stromal cells were also labeled with strong GFP at the end of the pulse phase. Secondly, we examined whether limbal epithelial cells expressed GFP without doxycycline induction. Corneal and limbal epithelial cells did not express GFP without doxycycline induction (Figure 2A,B), indicating that there were no false positive GFP signals due to leakage of GFP expression in our targeted tissue (limbal epithelium). Corneal and limbal stromal cells showed leakage of weak GFP expression without doxycycline induction (Figure 2B, not obvious under a low magnification in Figure 2A). Thirdly, we examined whether the label-retaining GFP^+^ cells contained LSCs with three methods. Method 1: under normal homeostasis during the chase phase, we observed that all limbal epithelial cells were GFP^+^ cells at the beginning of the chase phase (“Week 4” in Figure 2C), which gradually disappeared during the 4-week chase phase (Figure 2C). Only clusters of GFP^+^ cells remained in corneal and limbal epithelium (Figure 2A,C), suggesting that some corneal and limbal epithelial cells divided to support normal homeostasis and gradually lost their GFP label, and the more quiescent epithelial cells retained their GFP label with less cell division. Method 2: the label-retaining GFP^+^ cells in the limbal epithelium colocalized with limbal stem/progenitor cell markers cytokeratin 14 (K14) and p63 under normal homeostasis (“Unwounded” in Figure 2D) and under wounded conditions (“Wounded” in Figure 2D), indicating that the label-retaining cells showed a progenitor phenotype of epithelial cells. Method 3: under normal homeostasis, the GFP^+^ cells showed a spiral pattern on cornea at the end of the chase phase. When the corneal epithelium was mechanically removed, the limbal epithelium proliferated and healed the corneal epithelial wound, showing a GFP^+^ migration path from the limbus to the central cornea (Figure 3). This indicates that the limbal epithelial GFP^+^ cells responded to the corneal epithelial wound by cell proliferation and migration to heal the epithelial wound, which is a character of limbal stem/progenitor cells. The results showed that the GFP–H2B mice were a suitable model to study LSCs due to the absence of false-negative and false-positive GFP signals in our targeted tissue (limbal epithelium) and the inclusion of limbal stem/progenitor cells in the label-retaining GFP^+^ cells.

### 3.2. The Repetitive Wound Healing Procedure Led to a Slower Epithelial Wound Closure

To avoid any damage to LSCs, the wounding was restricted to the central corneal epithelium, leaving the peripheral corneal epithelium untouched (Figure 4A). Due to the regenerative capacity of the progenitor cells in the remaining peripheral corneal epithelium [24,25], two rounds of wound healing were performed to activate more limbal stem cells during the wound healing procedure. Corneal epithelial cells were gently removed by Algerbrush with minimal damage to the underlying stromal cells, and the epithelial wound healed without obvious structural distortion after two rounds of wounding procedures (Figure 4C). We observed that the closure of epithelial wounds, revealed by fluorescein staining, was slower after the 2nd wound compared to the 1st wound (Figure 4A,B). After the 1st wound, which removed around 49% of the corneal epithelium, the wound re-epithelized quickly, and only 6% of the corneal surface was not covered by functional corneal epithelial cells after 1 day. During the 2nd wound, a significantly smaller wounding area (around 38% of the corneal surface) was created, but around 14% of the corneal surface was not re-epithelized after 1 day, indicating a slower epithelial wound closure after the 2nd wound, which might be due to a partial exhaustion of limbal stem/progenitor cells during the repetitive wound healing procedure.

### 3.3. Single-Cell RNA Sequencing and Data Analysis

The two rounds of wound-healing procedures were performed on the OD eye of each mouse (referred to as “eyes/corneas under wounded condition” in the context) and the OS eye served as the unwounded control eye to minimize the variations according to individual, age, and sex. Limbal epithelial cells (LECs) and anterior limbal stromal cells were isolated from unwounded eyes and wounded eyes (pooled cells from 8 eyes for each sample), respectively, using an optimized protocol [23], followed by flow cytometry to obtain the label-retaining GFP^+^ cells for single-cell RNA sequencing. We obtained 11 clusters from 3136 cells in unwounded eyes (Figure 5A) and 12 clusters from 4967 cells in wounded eyes (Figure 5B). The mean reads per cell were 62,643 and 38,900 for unwounded eyes and wounded eyes, respectively. The median genes per cell were 3168 and 2834 for unwounded eyes and wounded eyes, respectively. The characteristic markers for each cluster were calculated by Seurat. To find the correlation among clusters from unwounded and wounded eyes, the clusters from both samples were ranked together into a dendrogram by hierarchical clustering.

The nature of each cluster was identified by their characteristic markers, which was calculated by Seurat as the differentially expressed genes in each cluster compared to the rest of the clusters. Cytokeratins are epithelial-specific intermediate filaments [26]. Based on the expression of cytokeratins (*Krt* family including *Krt12*, *Krt13*, *Krt14*, *Krt15*, *Krt17*, *Krt19*, etc.), we segregated the clusters into epithelial clusters (8 clusters for unwounded eyes and 8 clusters for wounded eyes) and non-epithelial clusters (3 clusters for unwounded eyes and 4 for wounded eyes). The non-epithelial clusters fell into three categories: stromal cells (identified by markers including *Col1a1* [27], *Col1a2* [27], *Dcn* [28], and *Cd34* [29]), T cells (identified by markers including *Trdc* [30], *Cd3g* [31], and *Trbc1* [32]), and macrophages (identified by markers including *Cxcl2* [33], *Ccl4* [34], and *Cd14* [35]). The epithelial clusters formed three major trajectory branches in the cluster dendrogram, designated as “Branch 1”, “Branch 2”, and “Branch 3” (Figure 5C).

In the dendrogram, the expression profiles of different clusters were compared and ranked so that fewer branches and shorter branches between two clusters indicate a lower difference in gene expression (Figure 5C). For the two clusters which were connected directly by a bracket without further branches (or the “smallest unit” of branches), if they were from the unwounded and wounded eyes, respectively, they shared very similar characteristic molecular markers and were thus considered as the same type of cells under normal homeostasis and under wounded conditions, respectively. For example, EC7 and EC15 were considered as the same cell type (“putative LSC1” in the 3rd column of Figure 6A). For the total 16 epithelial clusters (8 clusters for unwounded eyes and 8 clusters for wounded eyes), it yielded 11 cell types, of which 5 cell types were present in both unwounded and wounded corneas (black font in the 3rd column of Figure 6A), 3 cell types were present in unwounded corneas only (green font in the 3rd column of Figure 6A), and 3 cell types were present in wounded corneas only (red font in the 3rd column of Figure 6A).

We further determined the differentiation stages of the 11 epithelial cell types based on the expression of maturation markers and their position in the dendrogram. We did not employ the putative limbal/progenitor markers published previously to determine the cell type and differentiation stages in data analysis, because multiple stem and progenitor cell types were involved in the study in which the expression level of these putative limbal/progenitor makers are not conclusive yet. Although the definitive markers for LSCs and LPCs are not conclusive yet, the maturation markers for corneal epithelium and conjunctival epithelium are widely accepted, including *Krt12/Slurp1* [36,37] as the maturation markers for corneal epithelium and *Krt13/Krt8/Krt19* [38,39] as the maturation markers for conjunctival epithelium. Different maturation markers identifying the same type of cells showed a consistent trend in our data. Therefore, we picked *Krt12* and *Krt13*, which are the most widely used markers for corneal and conjunctival epithelium, respectively, as a criterion. They are shown in Figure 5 to evaluate the level of differentiation for the epithelial cell types. We observed a natural segregation on expression levels of the maturation markers in the epithelial cell types. The *Krt12* expression fell into 3 main ranges in epithelial cell clusters from the unwounded eyes: “<0.9” (same as the non-epithelial clusters), “5.5–20”, and “>180”, based on which we labelled them with a corneal differentiation score according to which “1” was “least differentiated”, “2” was “medium differentiated”, and “3” was “most differentiated”. The *Krt13* expression fell into 2 ranges in epithelial clusters: “<0.8” and “>9”, based on which we labelled them with a conjunctival differentiation score according to which “1” was “undifferentiated”; “2” was “more differentiated” (Figure 5). Based on the differentiation score, all cell types in Branch 1 were least differentiated under normal homeostasis and maintained their undifferentiated status upon wounding. All cell types in Branch 2 were differentiated at a medium-to-high level, and the cell types in Branch 3 showed a classic differentiation trajectory from least differentiated cell types (putative LSCs) to partially differentiated cell types under normal homeostasis and upon wounding. The characteristic markers of each cell type were shown in Figure 6A.

Among the three epithelial branches, Branch 3 is the only one that showed the differentiation trajectory from undifferentiated cells to partially differentiated cells under normal homeostasis and upon wounding. Therefore, we designated the undifferentiated cell types in Branch 3 as two putative limbal stem cell types (“putative LSC1” for EC7/15 and “putative LSC2” for EC8/16). Both putative LSC types had a comparable level on the high expression of epithelial progenitor markers (*Krt14*, *Krt15*) and the low expression of maturation markers (*Krt12*, *Krt13*), indicating that they were at a comparable undifferentiated status. We observed that “putative LSC2” showed a decreased percentage in cell number in the isolated label-retaining epithelial cells (only epithelial clusters were counted) after wounding (20% in unwounded corneas V.S. 11% in wounded corneas) (Figure 5C). The reduced percentage of “putative LSC2” might be due to the activated cell proliferation after wounding, which led to the loss of GFP label in some of the active dividing cells, indicating that the “putative LSC2” may be the “proliferative stem cells” to heal the epithelial wound, thus designated as “putative active LSCs”. In contrast, “putative LSC1” did not show a decrease in cell percentage upon wounding (18% in unwounded corneas V.S. 22% in wounded corneas) (Figure 5C), indicating that they might be at a quiescent status upon corneal epithelial wounding and are thus designated as “putative quiescent LSCs”. The spatial expression patterns of the two putative LSC types were examined by immunohistochemistry using their characteristic markers on frozen human corneal sections. *Edn2* and *Adm* were screened out as the marker for “putative quiescent LSCs” and *Fmo2* as the marker for “putative active LSCs”. EDN2, ADM, and FMO2 showed a distinctive high expression in discrete cells located at or near the basal limbal epithelium where LSCs are thought to reside (pointed at by white arrows, and pointed areas were enlarged at the bottom right corner in Figure 6B). The expression of *Edn2* and *Adm* was largely reduced after wounding (*Edn2* “29” and *Adm* “3.9” in EC7 compared to *Edn2* “13” and *Adm* “1.0” in EC15), while the expression of *Fmo2* was slightly reduced after wounding (“8.9” in EC8 compared to “6.8” in EC16). The partially differentiated cell types with a higher expression of maturation markers in Branch 3 were identified as putative limbal progenitor cells (LPCs) (putative “LPC1” for EC5, “LPC2” for EC6, and “LPC3” for EC14) (Figure 6A). EC5 had a significantly higher expression of *Mki67* and minichromosomal maintenance (*MCM*) family genes (cell proliferation markers), suggesting a proliferative nature of this cell type.

The cell types in Branch 2 were partially differentiated cells towards corneal epithelium and/or conjunctival epithelium. The cell types of EC3 and EC10 were designated as “differentiating LPC4” (LPC: limbal progenitor cells”) due to their higher *Krt12* expression than the LPC1/2/3 and were thus in a further “differentiating” status (Figure 6A). The cell types of EC4 and EC11 were designated as conjunctival progenitor cells (“CjPC”) due to their high expression of *Krt13* (Figure 6A). The cell types of EC12 and EC13 were from the wounded corneas only and were thus designated as “LPC5” and “LPC6” upon wounding, respectively (Figure 6A). EC13 had a characteristic high expression of *Mki67* and *MCM* family genes (cell proliferation markers) compared to the other clusters from the wounded corneas, indicating that EC13 is a proliferating epithelial progenitor cell after wounding.

Branch 1 contained two cell types, both of which showed undifferentiated status by low expression of the maturation markers *Krt12/Krt13* under normal homeostasis. The cell type of EC1 was only present in normal homeostasis but absent upon wounding and was thus not included as an LSC candidate. The cell type of EC2 and EC9 showed undifferentiated status under both normal homeostasis and wounding. However, it lacked the differentiation trajectory into more differentiated progenitor cells. In addition, its expression of epithelial progenitor markers (*Krt14*, *Krt15*) was lower than the two putative LSC types and the three putative LPC types in “Branch 3” but higher than the “putative differentiating LPC4” and “putative CjPC” in “Branch 2” (Figure 6A). The lack of differentiation trajectory and the moderate expression level on epithelial progenitor markers indicates that it may not be an LSC candidate. The two cell types in Branch 1 were located in a separate branch distal from the rest of the epithelial and non-epithelial clusters in the dendrogram. Their expression pattern and molecular markers did not give information on the potential source or lineage of the cells. For example, both cell types showed a low expression on the hematopoietic stem and progenitor cell markers including *Cd34*, *Cd44*, *Cd14*, and *Cd19* [40,41]. Therefore, they were designated as “putative stem/progenitor cells with unknown origin”.

We further examined the maturation markers whose expression was low in the putative LSCs, medium in the putative LPCs, and high in the differentiating LPCs, and ranked them in a list from high to low in fold differences. Only the cell clusters from unwounded eyes were included to study the maturation markers because the expression level during wound healing might be transient and stage specific, which may not serve as a marker during normal homeostasis. Ten putative maturation markers on the top of the list were selected showing increasing expression levels along the differentiation trajectory: low expression in the putative LSC types (EC7/15, EC8/16), medium expression in the putative LPC types (EC5, EC6, EC14), and high expression in the putative differentiating LPC type (EC3/10) (Figure 7A). Their expression levels were further examined between corneal and limbal epithelial cells isolated from human corneas. Four out of the ten markers (*Aldh3*, *Slurp1*, *Tkt*, *Krt12*) showed a significantly higher expression in human corneal epithelium than in human limbal epithelium through real-time quantitative PCR (Figure 7B).

## 4. Discussions

Significant effort has been devoted to the study of LSCs and their difference from many research groups using different models and different research tools, including single-cell RNA sequencing. In this study, we employed three characteristics of LSCs to identify them and to study their behavior and expression: (1) LSCs are quiescent under normal homeostasis, (2) LSCs can differentiate into LPCs under normal homeostasis and under wounded condition, and (3) LSCs are undifferentiated cells with minimal expression of maturation markers. In this study, we used H2B–GFP transgenic mice to enrich LSCs in the label-retaining GFP^+^ cells. The label-retaining cells were collected from unwounded mouse corneas and from repetitively wounded mouse corneas, respectively, and were loaded for single-cell RNA sequencing. The cell clusters from the two samples (unwounded and wounded corneas) were analyzed together in a dendrogram for the differentiation trajectory. The epithelial clusters were identified by the expression of epithelial-specific cytokeratin genes and were segregated into “putative stem cells”, “putative progenitor cells”, and “putative differentiating progenitor cells” due to their distinctive increasing expression levels on maturation markers for the corneal and conjunctival epithelium. We observed a branch in the dendrogram (“Branch 3”) showing a classic differentiation trajectory from undifferentiated clusters to partially differentiated clusters under normal homeostasis and under wounded condition. Therefore, the two cell types of the undifferentiated clusters were designated as “putative LSC1” and “putative LSC2”, respectively. The differences between these two putative LSC types were studied. “Putative LSC1” and “putative LSC2” were at a comparable level of undifferentiated status because they showed a comparably high expression on the epithelial progenitor markers (*Krt14*, *Krt15*) and low expression on the maturation markers (*Krt12*, *Krt13*). Putative LSC2 showed a decreased percentage in cell number after repetitive wounding, which indicates that this type of cell was induced to divide upon wounding and was thus designated as “putative active LSCs”. This is consistent with our result on a delayed epithelial wound healing after the second wounding, which might be due to activated cell division and a partial exhaustion of the “putative active LSCs”. Putative LSC1 did not show this decreased percentage in cell number after wounding, indicating that they might stay in a relatively quiescent status upon wounding. They were thus designated as “putative quiescent LSCs”.

The molecular marker of “putative active LSCs” was *Fmo2*. FMO2, a dimethylaniline monooxygenase, showed a 74-fold higher expression in enriched human LSCs in culture than in differentiated corneal epithelial cells [42], suggesting that it may present a stem/progenitor phenotype. The expression of full-length FMO2 occurs only in 26% of African Americans but is absent in Caucasians and Asian-Americans; instead, they express a truncated nonfunctional FMO2 [43,44]. No report on corneal diseases has been found on the human population with truncated FMO2. Whether *Fmo2* plays a functional role or whether its role is monooxygenase-independent in corneal limbus is still unknown.

The molecular markers of “putative quiescent LSCs” were *Edn2* and *Adm*. *Edn2* is reported to be expressed in a striped pattern on mouse corneas under normal homeostasis, which mimics the striped pattern of limbal stem/progenitor cells [45,46]. It regulates angiogenesis and barrier functions in different tissues [47,48,49,50,51], suggesting that it may play a similar role in cornea. EDN2 was proposed to increase angiogenesis in granulosa cells from the ovary by inducing vascular endothelial growth factor (VEGF) expression [51]. Its role in regulating angiogenesis in the retina is more complicated. Under the pathological condition of ischemic-induced retinopathy in mice, inhibition of EDN2 receptors increased physiological angiogenesis and increased vascular repair but decreased pathological neovascularization [49]. Overexpression of *Edn2* in the retina using transgenic knockin mice showed arrested vascular growth and accumulation of macrophages in the subretinal space [48]. One-month administration of exogenous EDN2 in mice caused a breakdown of the blood–retinal barrier with increased vascular leakage and infiltrating macrophages [50]. Data from previous publications indicate that the fine balancing of Edn2 signaling is a key regulator in angiogenesis and the blood–retinal barrier. *Adm*, a vasodilator, plays an important role in endothelial vascular barrier functions (including blood vessel endothelial and lymphatic endothelial barrier and growth) and in epithelial barrier functions [52,53,54,55,56,57]. ADM administration decreased blood–brain barrier permeability, decreased trans-endothelial electrical resistance, and attenuated fluid-phase endocytosis in vitro [52], showing its function in regulating the blood vessel endothelial barrier. Administration of ADM in cultured dermal microlymphatic endothelial cells and in mouse tails stabilized the lymphatic endothelial barrier and regulated lymphatic permeability [54]. Disruption of ADM receptor complexes using inducible Cre transgenic mice led to corneal inflammation caused by acute and chronic lymphatic dysfunction with dilated lymphatics and disorganized lymphatic junctions [56,57], suggesting the function of *Adm* signaling in maintaining the lymphatic barrier. Administration of ADM has been shown to improve intestinal epithelial barrier function by decreasing inflammation, increasing tight junctions, and decreasing intestinal epithelial cell permeability in a rat model of ulcerative colitis [55]. The characteristic high expression of *Edn2* and *Adm* in “putative quiescent LSCs” may indicate their role in regulating barrier functions, including the angiogenic barrier, blood vessel endothelial barrier, lymphatic endothelial barrier, and limbal epithelial barrier functions in cornea. In addition, the data of single-cell RNA sequencing demonstrated that both *Edn2* and *Adm*, as the characteristic markers for “putative quiescent LSCs”, showed a decreased expression in wounded corneas compared to unwounded corneas in this cell type (*Edn2*: “29” in EC7 V.S. “13” in EC15, *Adm*: “3.9” in EC7 V.S. “1.0” in EC15), leading to a hypothesis that “putative quiescent LSCs” may regulate barrier functions by a high expression of the markers under normal homeostasis to form the barrier and a decreased expression of the markers under wound healing to allow a temporary leakage of the barrier to facilitate wound healing.

Current data suggest that LSCs may contain at least two types of cells: one is quiescent upon corneal epithelial wounding, and the other is proliferative upon corneal epithelial wounding. Interestingly, a previous publication also reported the existence of two LSC populations in mice: one quiescent population (qLSC) at the outer limbus and one active population (aLSC) at the inner limbus [22]. Our “putative active LSCs” and “putative quiescent LSCs” may partially overlap with their “aLSCs” and “qLSCs” because the markers were partially shared (*Atf3* for “aLSCs” and “putative quiescent LSCs”, *Ifitm3* for “qLSCs” and “putative active LSCs”), although the rest of the cell markers were different and cell behavior was different upon wounding. Our data are also consistent with previous reports on the co-existence of quiescent stem cells and active stem cells in different epithelial and non-epithelial tissues, including cornea, hair follicle, gut, and bone marrow [22,58].

Many more questions are yet to be answered regarding the two types of putative LSCs. Are “putative quiescent LSCs” and “putative active LSCs” interconvertible? If yes, what factors promote the conversion? Do they have differences in regenerative capacity in vitro? Can we promote the in vitro propagation of LSCs by enriching or purifying a specific type of LSCs before culture? What mechanisms do the “putative quiescent LSCs” regulate the barrier functions? Do the “putative quiescent LSCs” have additional functions other than the potential barrier function? Is it possible that they regulate long-term epithelial regeneration and are not responsible for short-term wound healing? Will the markers be useful for diagnosing LSC deficiency? Will the markers of different LSC types be able to distinguish different types of LSC deficiency (e.g., loss of barrier function vs incapacity to heal epithelial wound)?

In addition to the putative LSCs, different types of putative LPCs were screened out from our analysis. Under normal homeostasis, the “putative LPC1” was proliferative, which may contribute to the replenishment of the lost corneal epithelial cells due to normal wear and tear; the “putative LPC2” had a high expression of its marker *Ler5*, which is a transcription factor regulating heat shock response, thermal stress, and ionizing radiation survival [59,60,61,62]. Both putative LPC1 and LPC2 disappeared after repetitive wounding, indicating that they might be exhausted during wounding healing. The “putative differentiating LPC4” is at a further differentiating status compared to other putative LPC types due to a high expression of maturation markers (*Krt12*, *Slup1*, etc.). Its characteristic marker is *Lypd2*, which has been shown to be highly expressed in superficial limbal epithelial cells while lowly expressed in corneal epithelium and basal and suprabasal limbal epithelium [19], indicating that the “putative differentiating LPC4” may be located at the superficial limbal epithelium. Under wounded conditions, the putative LSCs differentiate into the “putative LPC3” with the marker *Rdh10*. *Rdh10* plays an important role in normal eye development, and *Rdh10*-dificient mice lack cornea [63], indicating that it may represent an important transient progenitor cell type during development and during wound healing to generate corneal epithelium.

Ten markers were selected as maturation markers from our single-cell RNA sequencing data, of which four showed a higher expression in human corneal epithelium than human limbal epithelium. All four markers (*Aldh3*, *Slurp1*, *Tkt*, *and Krt12*) have been reported previously for their abundant expression in corneal epithelium and served as maturation markers for corneal epithelium [36,37,64,65]. Our data further confirmed that their expression levels were consistent with each other and correlated to different stages of differentiation.

In conclusion, this study provides the atlas of different epithelial and non-epithelial cell populations at limbus under normal homeostasis and during wound healing. The discovery of different types of LSCs and LPCs under normal homeostasis and upon wounding may provide valuable information for understanding the dynamics of corneal epithelial regeneration.

## Figures and Tables

**Figure 1 cells-11-01983-f001:**
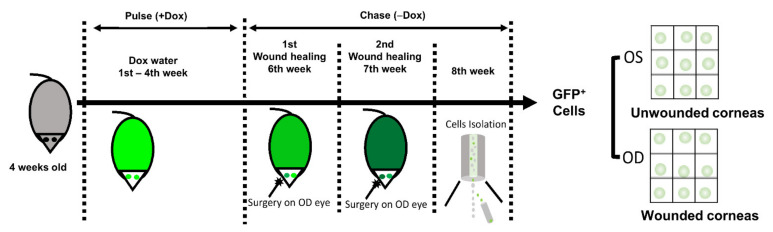
Experimental design of the wound-healing procedure performed on H2B-GFP mice to obtain the label-retaining cells in limbal epithelium. The 4-week-old H2B-GFP mice were fed with doxycycline (2 mg/mL) in sucrose water for 4 weeks during the pulse phase, followed by 2 rounds of corneal epithelial wound-healing procedure during the subsequent 4-week chase phase. At the end of the 8 weeks, unwounded eyes (OS eyes) and wounded eyes (OD eyes) were collected and their limbal epithelial and anterior stromal cells were collected respectively for flow cytometry to obtain the label-retaining GFP+ cells for single-cell RNA sequencing.

**Figure 2 cells-11-01983-f002:**
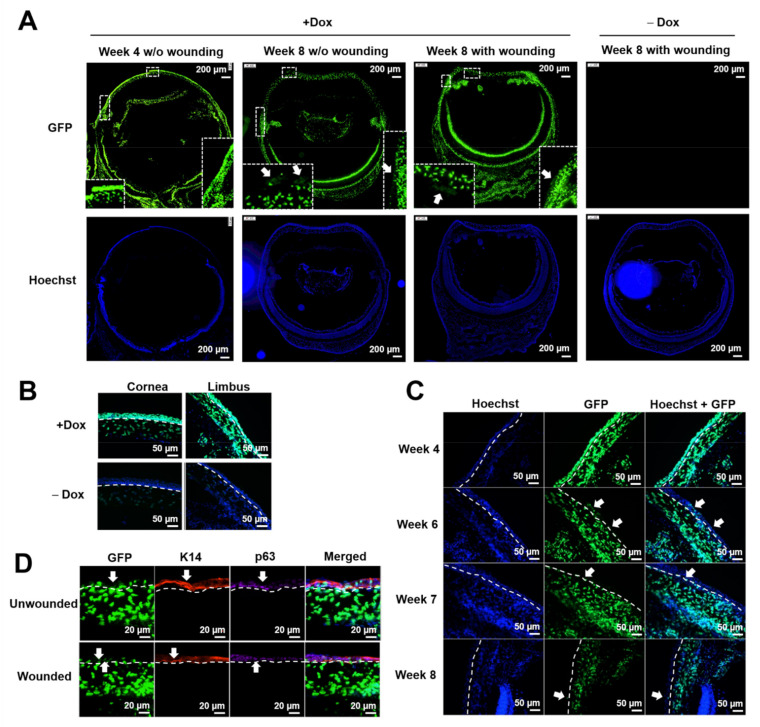
Validation of the label-retaining transgenic mouse model for the study of LSCs. (**A**) Representative pictures showing whole-eye cross sections of the H2B-GFP mice with or without doxycycline induction. Corneal and limbal epithelial and stromal cells were labeled with strong GFP at the beginning of chase phase (“Week 4”). At the end of chase phase (“Week 8”), majority of corneal and limbal epithelial cells lost their GFP label and the label retaining cells located as discrete cell clusters on corneal and limbal epithelium under normal homeostasis (“Week 8 w/o wounding”) and upon wounding (“Week 8 with wounding”). The eye cross section at the low magnification did not show obvious GFP signal without doxycycline induction at the end of chase phase with repetitive wounding. The enlarged corneal area was shown at the bottom left of each picture, and the enlarged limbal area was shown at the bottom right of each picture. White arrows pointed to the label-retaining GFP^+^ corneal and limbal epithelial cells. (**B**) Representative pictures at a higher magnification showing corneal and limbal epithelium and stroma were labeled with strong GFP at the beginning of chase phase (“Week 4”) with doxycycline induction. Corneal and limbal epithelium were GFP negative without doxycycline induction. Corneal and limbal stromal cells were GFP weak-positive without doxycycline induction. White dashed lines indicate the separation of epithelial cells and stromal cells. (**C**) Label-retaining cells located at discrete limbal epithelial cells during the 4-week chase phase. White arrows indicated the limbal epithelial label-retaining cells. White dashed lines showed the separation of epithelial cells and stromal cells. (**D**) The label-retaining cells showed a high expression of stem/progenitor markers cytokeratin 14 (K14) and p63 at limbus. Wounded corneas were collected at the end of repetitive wound healing procedure (13 days after the 2nd wounding). White arrows indicated the co-localization of the label-retaining cells with K14+ or p63-bright expression. White dashed lines showed the separation of epithelial cells and stromal cells.

**Figure 3 cells-11-01983-f003:**
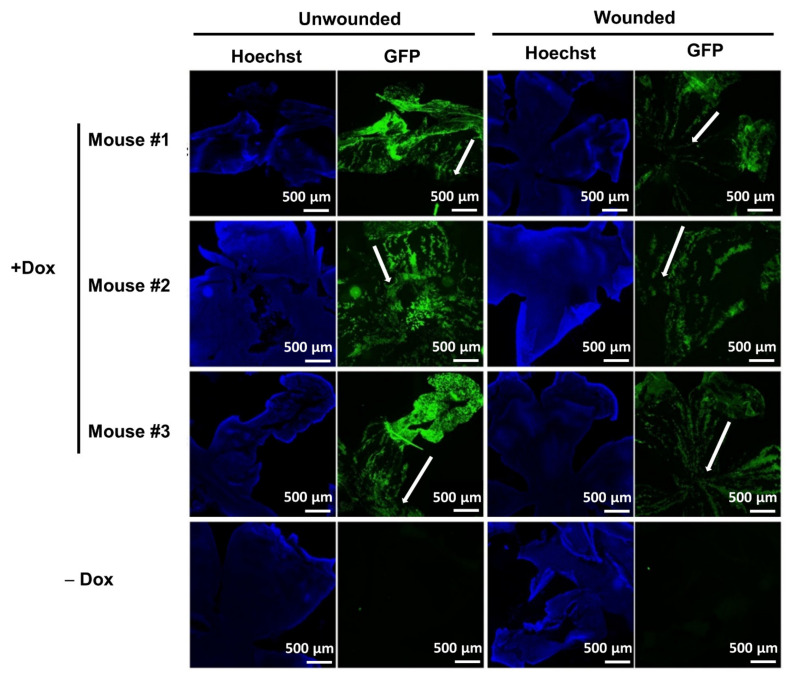
The label-retaining cells showed a spiral pattern under normal homeostasis and showed the migration path to healed corneal epithelial wound after repetitive wounding. Unwounded (OS eyes) and wounded (OD eyes) corneas were collected at the end of the repetitive wound healing procedure (13 days after the 2nd wounding). White arrows indicate the GFP^+^ spiral pattern under normal homeostasis or the GFP^+^ migration path from limbus towards the central cornea upon wounding.

**Figure 4 cells-11-01983-f004:**
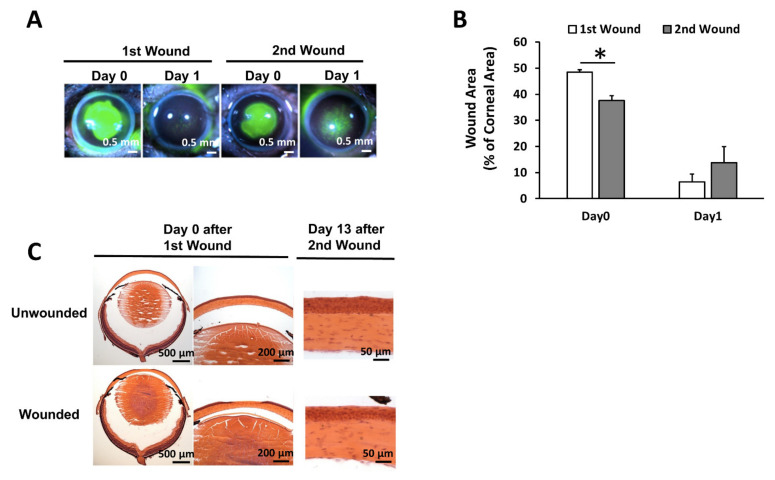
Repetitive wound healing procedure led to a delayed epithelial wound closure. (**A**) Representative pictures showing the corneal epithelial wound which was not covered by functional corneal epithelial cells after the first and second wounds using fluorescein staining. (**B**) Quantitation on the percentage of wound area after the first and second wounds. Percentage of wound area was calculated as the wound area revealed by fluorescein staining/the corneal area. *: *p* < 0.05. (**C**) Histological pictures showing that the repetitive epithelial wounding did not cause obvious damage to the stroma and the wound healed without an obvious scar. The tissue sections were stained with Hematoxylin and eosin (H&E).

**Figure 5 cells-11-01983-f005:**
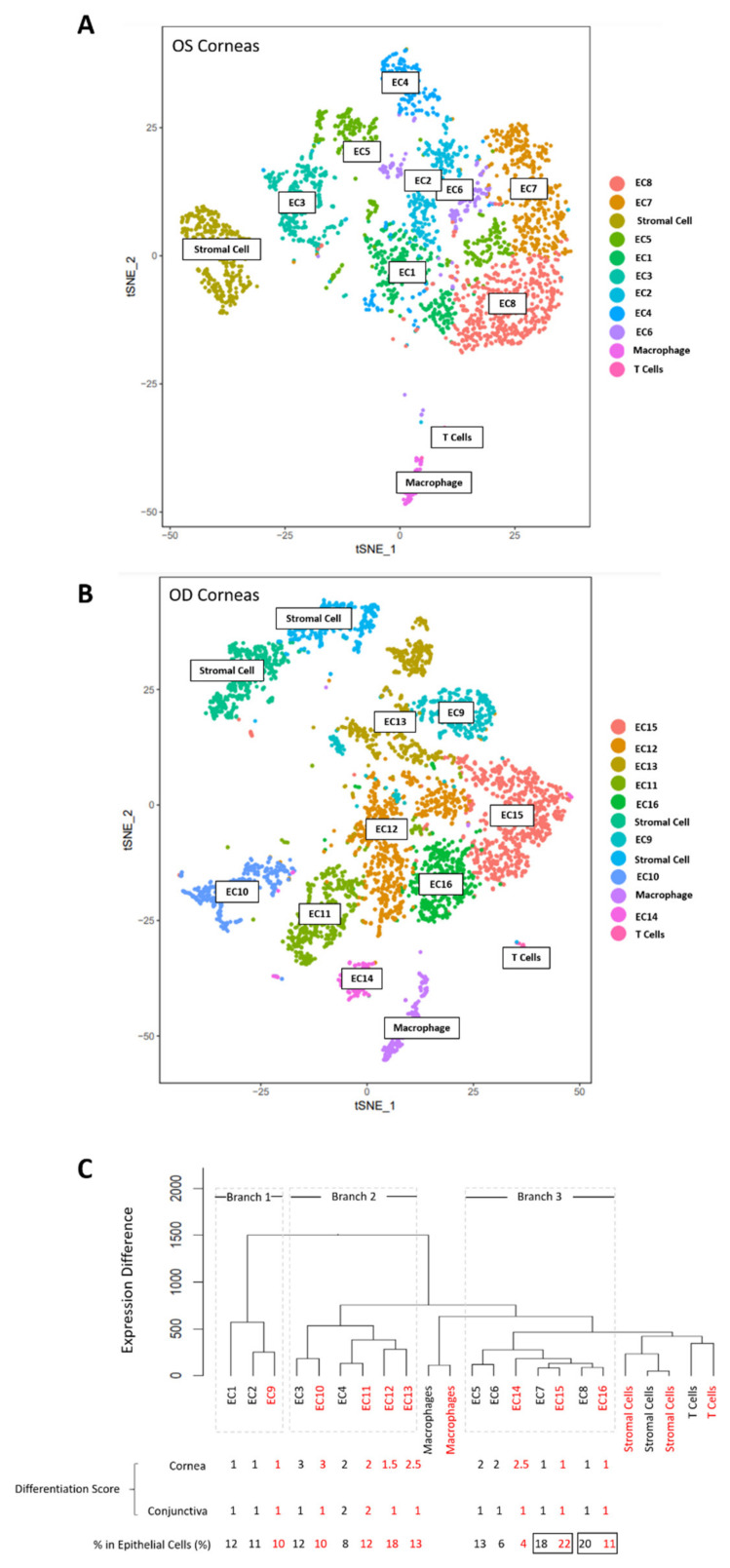
The heterogeneous cell populations at limbus revealed by single-cell RNA sequencing under normal homeostasis and upon wounding. (**A**) tSNE plot of the corneas under normal homeostasis. (**B**) tSNE plot of the corneas upon wounding. (**C**) Differentiation trajectory of the cell clusters from unwounded and wounded corneas. There were 3 branches of epithelial clusters in the dendrogram, designated as “Branch 1”, “Branch 2”, and “Branch 3”, respectively. EC: epithelial cluster. Differentiation score for cornea was based on the natural segregation of Krt12 expression: 1: <0.9, 2: 5.5–20, and 3: >180. Differentiation score for conjunctiva was based on the natural segregation of Krt13 expression: 1: <0.8 and 2: >9. The cell number of each EC during normal homeostasis and upon wounding were presented as the percentage of cell number in total epithelial cells (non-epithelial cells were not included). Black font indicates data of clusters under normal homeostasis. Red font indicates data of clusters upon wounding.

**Figure 6 cells-11-01983-f006:**
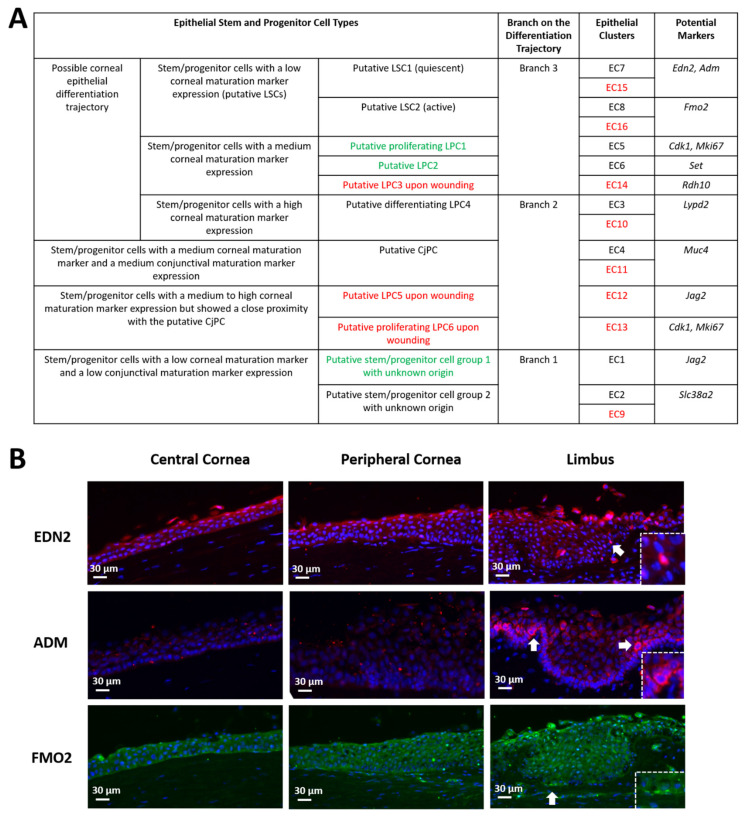
Putative LSC (limbal stem cell) and LPC (limbal progenitor cell) types revealed by single-cell RNA sequencing with characteristic markers. (**A**) The table showing different types of limbal epithelial stem and progenitor cells with characteristic markers. In the 3rd column, black font indicates cell types which were present under normal homeostasis and upon wounding; green font indicates cell types which were present under normal hemostasis but disappeared upon wounding; red font indicates cells types which were absent under normal homeostasis but appeared upon wounding. In the 5th column of “Epithelial Clusters”, black font indicates clusters under normal homeostasis, and red font indicates clusters upon wounding. (**B**) The two types of putative LSCs located as discrete cells at or near the basal limbal epithelium revealed by immunohistochemistry in human corneas using their characteristic markers. Arrows indicate the location of the cells with strong expression of the markers, which are shown as enlarged pictures in the squares at the bottom right corners.

**Figure 7 cells-11-01983-f007:**
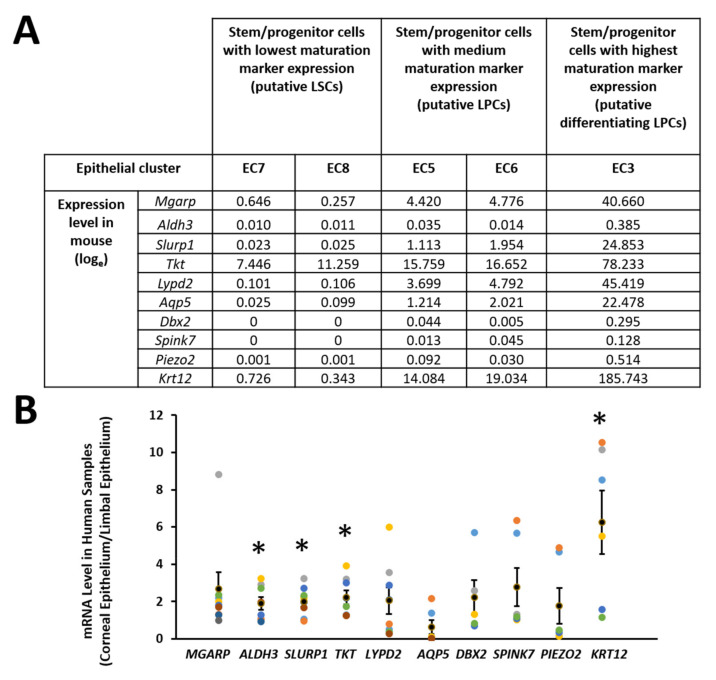
Maturation markers were selected for corneal epithelial differentiation. (**A**) Ten maturation markers were selected based on the increasing expression level along the differentiation trajectory (putative LSCs-putative LPCs-putative differentiating LPCs) revealed by single-cell RNA sequencing. (**B**) Four of the ten markers showed a higher expression in human corneal epithelial cells than those in human limbal epithelial cells revealed by quantitative real-time PCR. The Y-axis represents the fold changes which were calculated as gene expression in corneal epithelial cells/gene expression in limbal epithelial cells from human donors. *: *p* < 0.05. Dots with different colors indicate the samples from different donors.

**Table 1 cells-11-01983-t001:** Primers for Quantitative Real-Time PCR.

Gene	Primers
MGARP-F	CCCAGTGCTACAGTTGTGGT
MGARP-R	CCTCTGGGGTTGTTTCAGGG
ALDH3-F	TGATCCAGGAGCAGGAGCAG
ALDH3-R	GGACGTACACCACCTCCTCA
SLURP1-F	GTACCCCTTCAACCAGAGCC
SLURP1-R	GTCTCGGAAGCAGCAGAAGA
TKT-F	ACTTCGACAAGGCCAGCTAC
TKT-R	GCCCAGGCGATTGATGTCTA
LYPD2-F	CCGGGAGATAGTGTACCCT
LYPD2-R	AGTATTGCAGCAGGACACGG
AQP5-F	CCTGGCTGCCATCCTTTACTT
AQP5-R	AGGCTCATACGTGCCTTTGAT
DBX2-F	GTACTGGGACGTTGTGGCTT
DBX2-R	ACCCGCAGCAAATTCTCGAT
SPINK7-F	ATCCCCTGCCCCATCACATA
SPINK7-R	GCTCTCGGTACACAAGTGACA
PIEZO2-F	ACTTCCATGACCGGTTCCTT
PIEZO2-R	GGGTGGGCCAGTCTGTAG
KRT12-F	CCAGGTGAGGTCAGCGTAGAA
KRT12-R	CCTCCAGGTTGCTGCTGATGAGC
GAPDH-F	ACCCAGAAGACTGTGGATGG
GAPDH-R	CAGTGAGCTTCCCGTTCAG

## Data Availability

The data generated in this study are available from the corresponding author upon a reasonable request.

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
