# Peer review of "Differentiation Trajectory of Limbal Stem and Progenitor Cells under Normal Homeostasis and upon Corneal Wounding"

_cells, 2022, doi:10.3390/cells11131983_

Round 1
Reviewer 1 Report
The authors set to identify heterogeneous corneolimbal cell populations under normal homeostasis and upon wounding using single-cell RNA sequencing in a mouse model. They identified two putative LSC types showing a differential trajectory into limbal progenitor cell (LPC) types under normal homeostasis and during wound healing. They were designated as “putative active LSCs” and “putative quiescent LSCs”, respectively, because the former actively divided upon wounding but the latter type remained quiescent upon wounding. The authors speculate that the “putative quiescent LSCs” contribute to a barrier function because its characteristic markers regulate vascular and epithelial barrier and growth. Different types of LPCs at different proliferative status were identified in unwounded and wounded corneas with distinctive markers. Four maturation markers (Aldh3, Slurp1, Tkt, and Krt12) showed increasing expression from putative LSCs to LPCs. Thus, the study identified two different types of putative LSCs and several types of putative LPCs under normal homeostasis and upon wounding. The paper presents interesting data, especially with regard to using the technology of label retaining cell detection. However, the use of arbitrary markers to identify cell lineage and differentiation state seriously dampens the enthusiasm towards this manuscript.
This reviewer has the following significant concerns about the paper:
- Introduction:
Line 52: p63alpha should be replaced by deltaNp63alpha.
- Methods:
Please include catalog numbers of all antibodies used.
Please change euthanization to euthanasia.
The method of limbal cell isolation is unclear. The trephine at 1.5 mm would be inside the central cornea. How was the outer part demarcated and then freed from conjunctiva after sheet isolation? The same concerns human corneas. The presence of conjunctival cells needs to be controlled/determined.
Please state what cells were labeled in the cornea and in other organs. The H2B is not cornea-specific or epithelium-specific.
Please justify the inclusion of the studied genes.
Significant p value should be less than 0.05 (<0.05).
- Results:
The figure legends are absent.
The tiff files have a gray background. Please remove.
Due to slow cycling if any of the stromal cells, they should also retain the label and may contaminate the isolates.
It is unclear why two rounds of wounding were performed.
The schematic shows that the scRNA-seq was performed on healed corneas at least a week after wounding. At that time the mouse corneas would become quiescent. How can one attribute wound-healing related functions to LSCs in this situation? Why not check these populations when the wounds are still healing or just have healed? At what time after wounding the “wounded” corneal pictures were obtained?
On sections, please mark the epithelium and stroma.
Figure 4 does not add to the study and may be removed.
How was the scRNA-seq performed: on single or on pooled corneas?
The term “under wounded condition” is vague. Please explain in detail.
It is completely unclear how the markers characterizing different clusters were assigned to a cell type. The strategy has to be presented in detail. As is, it looks rather arbitrary, especially that the authors did not use previously published markers associated with specific cell types. This appears as a serious weakness. In fact, apart from ADM, other stainings are not even limbal-specific (Fig. 6).
The analysis of the literature appears to be rather superficial and the conclusions look very speculative. Pertinent literature is not well cited.
Reviewer 2 Report
- In Figure 7B, the error bar of AQP5 gene qPCR results is too large. In order to eliminate machine faults, it is recommended to repeat this experiment, and it is best to make scatter diagram after repeated for several times.
- In Figure 2A upper panel and Figure 4A, please add scale bar.
- Please add Figure legends.
- In line 115, 118, 168 and 171, please indicate dilution ratio of the antibodies.
- Between line 188 to 193, please specify the qPCR running program.
- In line 198, “P ≤ 0.5 was considered statistically significant.”, however, P<0.05 was considered statistically significant. Please indicate whether it is an input error or data analysis fault.
- It is recommended that the language be modified and polished by professionals. For example, in line 170 and 182, the degrees Celsius sign is inconsistent; in line 134 and 166, one have Space between number and time unit, while the other does not; in line 425, it looks like two Spaces were entered after the period.
Reviewer 3 Report
The paper titled "Differentiation Trajectory Of Limbal Stem And Progenitor Cells Under Normal Homeostasis And Upon Corneal Wounding" aims to identify various cells populations at the limbus under normal and wounding conditions. Corneal repair is an important issue waiting for the optimal solution, therefore this research study is interesting not only for researchers, but also for clinicians dealing with problems in regenerating the affected corneas by various mechanisms of injury. The methodology of the study is described in a detailed manner, the results are clearly presented and discussed relatively to already published data, outlining the questions that are still to be answered. The authors offer the proof of discovering different types of LSCs and LPCs under normal homeostasis and upon wounding which opens a path for understanding the dynamic of corneal epithelial regeneration.
Round 2
Reviewer 1 Report
The paper has been improved but significant concerns still remain about most of the important points of the paper. Specifically,
1. The genes that the authors studied by RT-PCR have been poorly justified again; the only thing that was included was “ten genes selected from the single-cell RNA sequencing data as putative maturation markers”. No references are provided and these markers do not seem to belong to stem/progenitor markers.
2. The authors describe the wounding as follows: “Central corneal epithelium was carefully removed by Algerbrush, and the complete removal of epithelium was confirmed by fluorescein staining”. In this condition, their justification of two wounds as engaging more progenitor cells does not stand criticism, as only limbal cells would participate in the healing process due to removal of entire epithelium. Further, the fluorescein staining may only indicate the size of the wound but not really control the “complete removal of epithelium”. The allusion to the Majo paper that has been refuted by three independent groups using much more sophisticated methods does not add to the justification.
3. The authors were asked to justify analyzing “wounded” corneas at least a week after wounding when the healing process would be long complete. Their explanation is that “The reason is that if the cells were collected during the active healing phase, they would have a significant difference on the gene expression profile compared to their quiescent status”. However, wasn’t it the goal of comparing wounded and unwounded states that the authors discuss abundantly? What is then the rationale behind wounding corneas at all? The authors themselves later said: “Corneas “under wounded condition” are the corneas that underwent the two rounds of wound-healing procedures”. But they have long healed. What is “under wounded condition” then? This explanation cannot be agreed with.
4. The explanation of figure 4 is rather expected and trivial. This reviewer is still convinced that it does not add to the story because the second wounding has not been properly explained.
5. “The characteristic markers for each cluster were obtained by Seurat based on its intrinsic mathematical formulas to calculate the difference of gene expressions among different clusters”. This explanation still does not explain at all why the pertinent “markers” were considered in the biological context. The assurance that “We observed a strong expression of EDN2 and FMO2 in a rare number of discrete basal limbal epithelial cells, which is consistent with the estimated rare population of limbal stem cells” is not supported by Fig. 6 pictures. As the negative controls are absent, it is unclear what is specific staining for these two proteins. Again, only ADM seems to be limbal-specific. The choice of markers, in the opinion of this reviewer remains arbitrary.
Author Response
We would like to thank the reviewer for his/her detailed review of our manuscript. The reviewer’s comments were addressed one by one as follows.
- The genes that the authors studied by RT-PCR have been poorly justified again; the only thing that was included was “ten genes selected from the single-cell RNA sequencing data as putative maturation markers”. No references are provided and these markers do not seem to belong to stem/progenitor markers. --- Please see the more detailed explanations in Results (page14, line 452-461) and Discussions (page 18, line 592-598) with references cited. These markers were selected as maturation markers, which are the opposite of stem/progenitor markers.
- The authors describe the wounding as follows: “Central corneal epithelium was carefully removed by Algerbrush, and the complete removal of epithelium was confirmed by fluorescein staining”. In this condition, their justification of two wounds as engaging more progenitor cells does not stand criticism, as only limbal cells would participate in the healing process due to removal of entire epithelium. Further, the fluorescein staining may only indicate the size of the wound but not really control the “complete removal of epithelium”. The allusion to the Majo paper that has been refuted by three independent groups using much more sophisticated methods does not add to the justification. --- Thank you for pointing out our mistake for the fluorescein staining. The “complete removal of epithelium” has been changed into “the size of wound” in page 2, line 95. We mentioned the “complete removal of epithelium” in the previous version, which did not mean “the complete removal of all corneal epithelium”; instead, what we meant was “the complete removal of all layers of central corneal epithelium” (because we found that if we removed the superficial layers only, fluorescein could not stain the tissue due to the remaining layers of epithelium). We apologize for the confusion and have corrected the phrase. Actually the “complete removal of all corneal epithelium” is technically impossible due to the lack of clear separation/edge between corneal epithelium and limbal epithelium. To avoid any damage to limbal stem cells (which is the target of our study), we left enough space between the wounding edge and the limbus, leaving a good amount of peripheral corneal epithelium untouched after wounding (as shown in Figure 4). The un-damaged peripheral corneal epithelium contained progenitor cells, which also contribute to corneal epithelial wound healing. That’s why we performed the repetitive wound healing procedure to engage more cells activated from limbal stem cells. The explanation has been improved in page 8, line 286-290.
- The authors were asked to justify analyzing “wounded” corneas at least a week after wounding when the healing process would be long complete. Their explanation is that “The reason is that if the cells were collected during the active healing phase, they would have a significant difference on the gene expression profile compared to their quiescent status”. However, wasn’t it the goal of comparing wounded and unwounded states that the authors discuss abundantly? What is then the rationale behind wounding corneas at all? The authors themselves later said: “Corneas “under wounded condition” are the corneas that underwent the two rounds of wound-healing procedures”. But they have long healed. What is “under wounded condition” then? This explanation cannot be agreed with. --- We do not agree with the reviewer on this point. Wound closure does not equal to wound healing. When the wound was closed by epithelial coverage, the wound healing continues. Actually our data showed that after 13 days of the second wounding (which was about 9 days after wound closure indicated by fluorescein staining), the cells in wounded corneas were significantly different from the cells in unwounded corneas in terms of cell types and gene expression profiles.
The reviewer suggested to analyze the cells during an active wound healing stage, which could be done, but will risk the data analysis that we may not be able to link the cell subtype at the quiescent status with that under active wound repairing. The cell types present in our targeted tissue contained limbal stem cell subtypes and limbal progenitor cell subtypes, which are cell subtypes at different stages during the differentiation trajectory and the differences among them could be subtle. The differences of gene expression caused during an active wound healing stage (with active cell proliferation and migration, tissue inflammation, stimulated paracrine/chemokine signals, etc.) may override the subtle difference among the different cell subtypes and may lead to a failure to identifying and linking the cell types under unwounded and wounded conditions, thus not chosen for our study.
- The explanation of figure 4 is rather expected and trivial. This reviewer is still convinced that it does not add to the story because the second wounding has not been properly explained. --- As mentioned above, the reason of second wounding has been updated in page 8, line 286-290.
- “The characteristic markers for each cluster were obtained by Seurat based on its intrinsic mathematical formulas to calculate the difference of gene expressions among different clusters”. This explanation still does not explain at all why the pertinent “markers” were considered in the biological context. The assurance that “We observed a strong expression of EDN2 and FMO2 in a rare number of discrete basal limbal epithelial cells, which is consistent with the estimated rare population of limbal stem cells” is not supported by Fig. 6 pictures. As the negative controls are absent, it is unclear what is specific staining for these two proteins. Again, only ADM seems to be limbal-specific. The choice of markers, in the opinion of this reviewer remains arbitrary. --- The markers were selected using a non-biased method calculated mathematically by Seurat. We do not agree with the reviewer that the biological context should be considered during marker selection, because that would bring a subjective opinion from the beginning on marker selection affected by what the researchers think are LSCs. The fact is that we do not know what LSCs are and that was the purpose of this study: to find LSCs with markers which may or may not be identified previously and which may or may not has known biological functions. The negative controls (no primary antibody control) showed a clear background, which are attached below. We do not agree with the reviewer on the expression pattern of EDN2 and FMO2. Actually they showed a more specific pattern than ADM to label a rare population of basal limbal epithelial cells. In our opinion, EDN2 and FMO2 might be better and more specific markers for LSCs than ADM. EDN2 and FMO2 showed some moderate expression in central and peripheral cornea, which was lower in signal intensity compared to their expression in discrete limbal epithelial cells. This is expected because the expression in non-LSCs were not zero from our single-cell RNA sequencing data and we did expect a low-to-moderate level of gene expression in non-LSCs.
